

# Prochloraz induced alterations in the expression of mRNA in the reproductive system of male offspring mice

Junhe Hu, Chang Liu, Xianghui Zeng, Tao Tang, Zhi Zeng, Juan Wu, Xiansheng Tan, Qingxiang Dai and Chenzhong Jin

Hunan Provincial Key Laboratory of Pesticide Harmless Application, Loudi, Hunan Province, China
Department of Agriculture and Biotechnology, Hunan University of Humanities, Science and Technology, Loudi, Hunan Province, China

## ABSTRACT

Prochloraz is a widely used fungicide worldwide. It is classified as an endocrine disrupting pesticide that affects the reproductive system. This study aimed to examine the impact of exposure to prochloraz of male mice on the reproductive system of their offspring male mice. Male father mice were intragastrically administered different dosages of prochloraz (group MA: 0 mg/kg/day; MB: 53.33 mg/kg/day; MD:160 mg/kg/day). Then, the testicular average weight of male offspring in the dose groups was found to be significantly lower than those in the control group (MB:0.312g, MD:0.294g, and MA:0.355 g; $P < 0.05$). Additionally, the testicular coefficient index in the MB and MD groups was also lower than that of the control group. Secondly, we observed that there were significantly different expressed genes clustered in groups B and D, in contrast to the control. Finally, the findings demonstrated a significant alteration in the response of male mice reproductive relative genes to prochloraz invasion. Two genes (*Mt-nd6* and *Slc12a4*) were found to be involved in the regulation of sperm mitochondria function and six genes (*Greb1, Esrrb, Catsperb, Mospd2, Sohlh1* and *Specc1*) were closely linked to sperm functions and estrogen response. The study revealed a significant impact of prochloraz on the reproductive system of male mice, thereby supporting further investigation into the reproductive toxicological effects of the drug.

## INTRODUCTION

The extensive use of pesticides in agriculture has been associated with an increase in adverse reproductive effects for humans and wildlife due to chemical exposure. Reports on the impact of pesticides on wildlife reproduction are abundant. However, the effects of human exposure to these chemicals and their impact on the male reproductive system, including malformed sex organs, poor sperm quality, and increased incidence of testicular cancer, remains a topic of ongoing investigation. Recent research has suggested a causal link between human exposure to pesticides and issues like poor sperm quality, changes in testis epigenetics, male health, and an increased incidence of cryptorchidism (*Meng et al., 2018*; *Sifakis et al., 2017*). It is well-documented that

Corresponding author
Junhe Hu, 260477247@qq.com

many chemical pesticides can disrupt crucial endocrine processes, with some acting as antiandrogens (*Dellis & Papatsoris, 2019*; *Rider et al., 2008*; *Tang et al., 2016*; *Watermann et al., 2016*). Pesticides like prochloraz, vinclozolin, procymidone, linuron, and the phthalates DEHP (diethylhexyl phthalate) and DBP (dibutyl phthalate) are among the more well-researched antiandrogenic chemicals, showing an impact *in vitro* and in animal studies (*Birkhøj et al., 2004*; *Vinggaard et al., 2005*).

Prochloraz is an imidazole fungicide widely used in gardening and agriculture across Europe, Australia, Asia, and South America. It has been shown in screening studies to have multiple mechanisms of action. It acts as an antagonist to both androgen and estrogen receptors, serves as an agonist for the Ah (Aryl hydrocarbon) receptor, and inhibits aromatase activity. *In vivo*, prochloraz functions as an antiandrogen by reducing the weights of reproductive organs, influencing androgen-regulated gene expressions in the prostate, and increasing LH level (*Noriega et al., 2005*; *Vinggaard et al., 2006*). Imidazoles, such as prochloraz, act as fungicides or antimycotic drugs by inhibiting cytochrome P450-dependent 14a-demethylase activity, necessary for converting lanosterol to ergosterol (*Dang et al., 2016*; *Geißet al., 2017*). This inhibition is due to the imidazole moiety interacting strongly with the iron atom of cytochrome P450 (*Blystone et al., 2007*; *Zhang et al., 2020*). The binding is somewhat nonspecific, resulting in imidazole fungicides also inhibiting a wide range of other cytochrome P450-dependent enzymes, including key enzymes involved in the biosynthesis and metabolism of steroids, such as cytochrome (CYP) 19 aromatase (*Beijer et al., 2018*; *Hansen et al., 2017*).

Some chemical substances, such as prochloraz, have been shown to exhibit estrogen-like biological effects and impact testosterone production in male animals and humans, leading to various adverse effects on human health and male reproductive function (*Ma et al., 2021*). While there is a significant body of research on the effects of chemical pesticides on human health, particularly through experimental animal exposure to these hazardous substances (*Cescon, Chianese & Tavares, 2020*), the specific mechanism by which prochloraz impairs spermatogenesis and alters genomic expression in male offspring mice remains unclear. Three different doses of prochloraz were tested to investigate the impact of prochloraz on peri-postnatal development and reproductive toxicity: 0 mg/kg/day (Group MA), 53.33 mg/kg/day (Group MB), and 160 mg/kg/day (Group MD), representing 0, 1/30, and 1/10 of the median lethal dose of prochloraz (1600 mg/kg/day).

## MATERIALS AND METHODS

Portions of this text were previously published as part of a preprint (https://d197for5662m48.cloudfront.net/documents/publicationstatus/147948/preprint_pdf/71300a244e9692c72325d1212ba7e551.pdf).

### Animals and prochloraz preparation

Nine male Kunming mice (3 weeks old, weighing 25 ± 2 g) and eighteen healthy female Kunming mice (4 weeks old, weighing 20–22 g) were obtained from Hunan SJL Laboratory

Animal Co Ltd. The study was approved by the Institutional Ethics Committee of Hunan University of Humanities, Science and Technology (file number: 20211010). Mice were housed individually in cages in a room at 22–24 °C, relative humidity of 65%–80%, and a 12-hour light-dark cycle. They had access to food and water *ad libitum*. Prochloraz (CAS number 67747-09-5) was synthesized and characterized by Shanghai TITAN Technology (Tansoole Company, Shanghai, China) as a 98% pure brownish solid. It was dissolved in corn oil to prepare dosing solutions, which were assessed for accuracy and homogeneity by HPLC-MS analysis. The stability of prochloraz in corn oil was tested by analyzing samples stored at room temperature over 7 days. Fresh dosing solutions were prepared weekly.

## Experimental design and testis tissue collection

The experimental group was injected with a uniform suspension of prochloraz dissolved in solvent corn oil every three days without analgesia. The control group was injected with solvent corn oil. The volumes were measured based on the average of the two experimental groups. The mice were allowed to acclimate for three days and then nine male mice were randomly divided into three groups: control group (male mouse group A, MA: corn oil), low dose group (male mouse group B, MB: 53.33 mg/kg bw/day), and high dose group (male mouse group D, MD: 160.00 mg/kg bw/day). The daily injection amount was adjusted according to the mice's weight on the day of injection. Each group of three mice was intraperitoneally injected once a day at 3 p.m. for 14 days (two weeks). The male mice were stable for three days after the poisoning, during which the experimental female mice were injected with 10 IU of hemogonadotropin each, followed by 10 IU of chorionic gonadotropin 48 h later. They were then housed with the infected male mice in a 1:2 male-to-female ratio. Vaginal plugs were checked within 12–15 hours after pairing and the female mice were individually raised to breed offspring. Three male offspring mice from each group were selected for further feeding and experiments for 5 weeks, with free access to food and water. After 5 weeks, nine male offspring mice (three from each group) were further fed according to the experimental design. Male mice were euthanized by dislocation under sterile conditions, their testes were immediately excised, and adipose tissue was removed according to institutional guidelines. The testes were used for mRNA expression comparison among the three dosage groups using RNA-Seq technology. The experimental setup is illustrated in Fig. 1.

## RNA isolation and mRNA library construction

A total of nine left testis samples (three samples from each group) were selected for RNA isolation. Total RNA was isolated using Trizol Reagent (Cat#15596018; Invitrogen, Waltham, MA, USA) according to the manufacturer's instructions. Then RNA detection and quality control were carried out using an Agilent 2100 Bio-analyzer (Agilent Technologies Sweden AB) for the verification of RNA integrity. The mRNA Library construction is carried out as following.

The construction of the chain-specific library involves using dUTP instead of dTTP to mark the second strand of cDNA during synthesis. Uridine DNA glycosylase (UDG) was

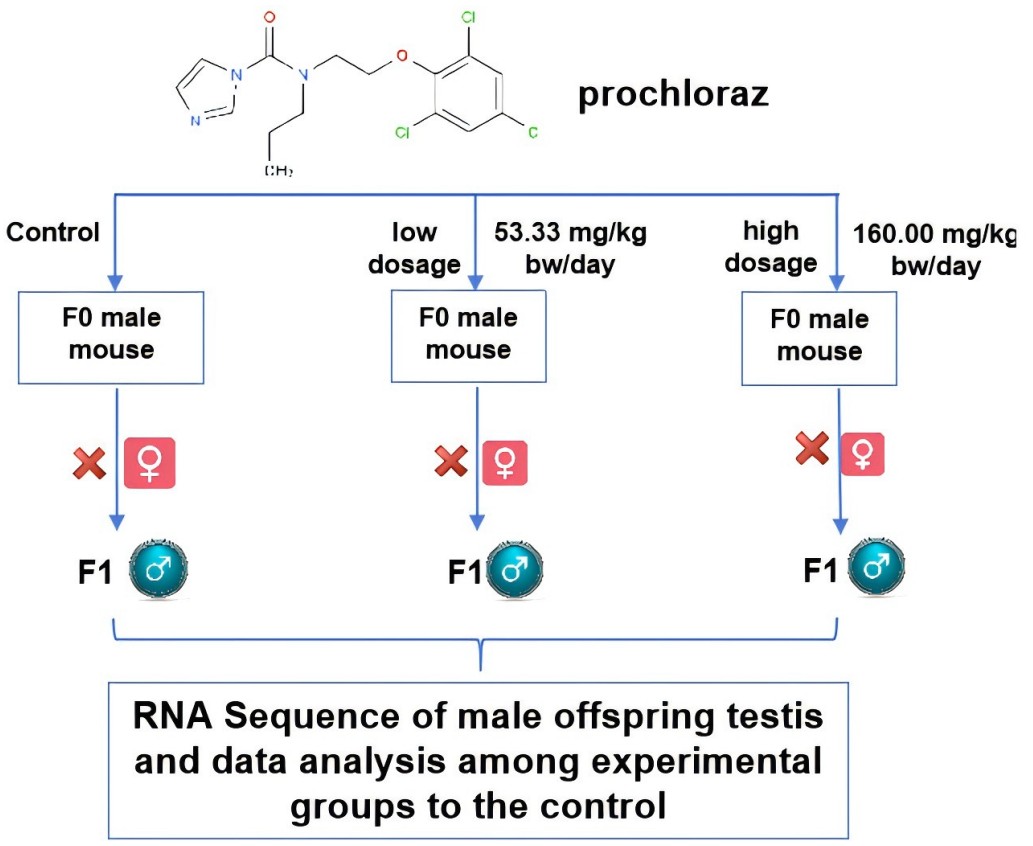

**Figure 1** **The schematic diagram of the experimental design.** The schematic diagram about the experiment is shown for researching the prochloraz's effects on male mouse F1 offspring reproductive system. The figure was made with PowerPoint software.

used to hydrolyze the chain before PCR enrichment, ensuring that the final sequencing data originates from the first strand of cDNA. Additionally, actinomycin D was included during the synthesis of the first strand of cDNA to inhibit the binding of reverse transcriptase and DNA template, thus preventing the synthesis of false negative chains and enhancing the specificity of sequencing data. The specific process includes segmenting the obtained mRNA fragments, synthesizing the first strand cDNA, synthesizing the second strand cDNA, preparing the ends of the cDNA library and adding a tail, connecting adapters, amplifying the library fragments through PCR enrichment, and purifying the library fragments using magnetic beads to obtain an ultra-micro rRNA detection library. Finally, the library's quality was assessed using the Bioptic Qsep100 analyzer to verify the size distribution aligns with the theoretical size. Following quality control of the mRNA library, the Novaseq's high-throughput sequencing platform and PE150 sequencing platform are employed for further analysis.

### Read alignment and RNA-seq data analysis

The read quality was estimated with the FASTQC program and the statistical power of this experimental design, calculated in RNASeqPower is 0.87. RNA Sequencing Data is

**Table 1  Prochloraz' effects on male mouse testicular weight growth and testicular coefficient index.** The different lowercase letters in the same line in the table indicate the significant difference at the level of $P < 0.05$, (x $\pm$ 4 s; n = 3); the testicular coefficient index is equal to the weight of testis divide the body weight.

| Measuring Index | Group MA (control group) | Group MB (low dosage group) | Group MD (high dosage group) |
|---|---|---|---|
| testicular weight (g) | $0.355 \pm 0.022a$ | $0.312 \pm 0.024b$ | $0.294 \pm 0.005b$ |
| testicular coefficient index | $0.0091 \pm 0.00084a$ | $0.0086 \pm 0.001a$ | $0.0085 \pm 0.0004a$ |

available at the National Genomics Data Center: CRA012977. Fastp software was used to control and preprocess the fastq data. Reads were aligned to the mouse Ensemble genome GRCm38 (https://www.gencodegenes.org/mouse/) using the Hisat2 aligner (v2.1.0) under parameters: "–rna-strandness RF". Differential gene expression analysis was performed using the DESeq2 R-package. Each experimental group replicates at three times. Finally, we applied a DEBseq-counts algorithm to filter the differentially expressed genes following FDR analysis using the criteria |log2FC | > 0.585 and $P < 0.05$ to choose the significantly expressed genes for further research.

Functional annotation was performed using the Database for Annotation, Visualization and Integrated Discovery (DAVID) v6.8 (https://david.ncifcrf.gov). Pathway analysis was carried out using annotated data downloaded from KEGG (https://www.genome.jp/kegg/).

## Statistical analysis

The back-spliced junction and linear mapped reads were merged and normalized per kilobase per million mapped reads (RPKM) to assess the mRNA expression levels across three distinct groups. Mean values and standard deviations were computed. Furthermore, various statistical analyses were conducted, including ANOVA analysis comparing treated groups to control groups. Statistical significance was defined as $P < 0.05$. In addition to the statistical tools integrated within bioinformatics software and resources, supplemental statistical analyses were carried out using GraphPad Prism (Version 5.00).

## RESULTS

### Effects of prochloraz on testicular weight and testicular coefficient in male offspring mice

Male mice experienced significant changes in their reproductive organs under the influence of antiandrogens. Table 1 presents the analysis of testicular mass and testicular coefficient in male mice. A notable difference in testicular weight was observed between the experimental groups and the control group ($P < 0.05$), while no significant variance was found in testicular weight between the high and low concentration groups ($P > 0.05$). Although the testicular coefficients of mice in both concentration groups were lower compared to the control group, the variance was not statistically significant ($P > 0.05$). This suggests that prochloraz plays a crucial role in male mouse testis development by influencing testicular weight but further research at the molecular level is required to elucidate the mechanism behind the inhibition of male mouse testis development in future experiments.
## Quality control and mapping of RNA-Seq data

To ensure the credibility of subsequent analysis results, the raw reads generated by the sequencing machine were evaluated and filtered. The objective of sequence filtering is to obtain high-quality clean reads suitable for further analysis. Fastp (https://github.com/OpenGene/fastp) was utilized for this purpose, which filters sequencing data by removing splices, N-containing bases, and low-quality bases, while also assessing overall data quality. These results provide a deeper insight into the sequencing data prior to analysis. The sequence quality control statistical results for the samples are presented in Tables S1 and S2.

## Different gene analysis among different groups

The distinct genes between the experimental and control groups must be identified and correlated with the phenotype to determine phenotypic gene expression. Different sequencing data types and grouping methods require specific screening algorithms and standardized procedures to ensure the effectiveness of data analysis. For this study, deseq2 was used, and significant differentially expressed genes were identified based on two criteria: $|log2FC| > 0.585$ and $P$ value $< 0.05$.

Cluster analysis was performed on the expressed genes to assess their relationship. The correlation of samples was determined based on the expression of selected differential genes. Typically, samples of the same type tend to cluster together, indicating that genes within the same cluster may share similar biological functions (Fig. 2). When compared to the control group, the low concentration treatment group exhibited a significant difference in RNA expression, with improved sample repeatability demonstrated through three repeated experiments (groups MB1, MB2, and MB3 as depicted in Fig. 3). Similarly, the high concentration treatment group also showed a notable difference in RNA expression compared to the control group, with good sample repeatability across three repeated experiments (groups MD1, MD2, and MD3 as shown in Fig. 3). Additionally, the RNA expression in the high concentration group was significantly different from that of the low concentration group, with consistent repeatability (Fig. 3). Notably, RNA expression within the same concentration groups tended to cluster together.

## Gene function and signal pathway analysis

The hierarchical structure of GO analysis organizes the mutual regulation and subordination among all GO terms into a database. By constructing a functional relationship network, the affected functional groups and the internal subordination of significant functions can be easily summarized. In this study, the significance of GO terms ($P$ value $< 0.01$) in the GO analysis of differential genes was used as the research focus to analyze the functional regulation and construct the functional regulation network. To accurately classify these genes, GO analysis assigns different functional classifications to significant different genes. The GO classification can describe gene function from various aspects and GO can be divided into three main groups: biological process (BP), molecular function (MF), and cellular component (CC). Based on the GO annotation of BP, MF, and CC, all the GO terms involved in genes were obtained in our experimental results.

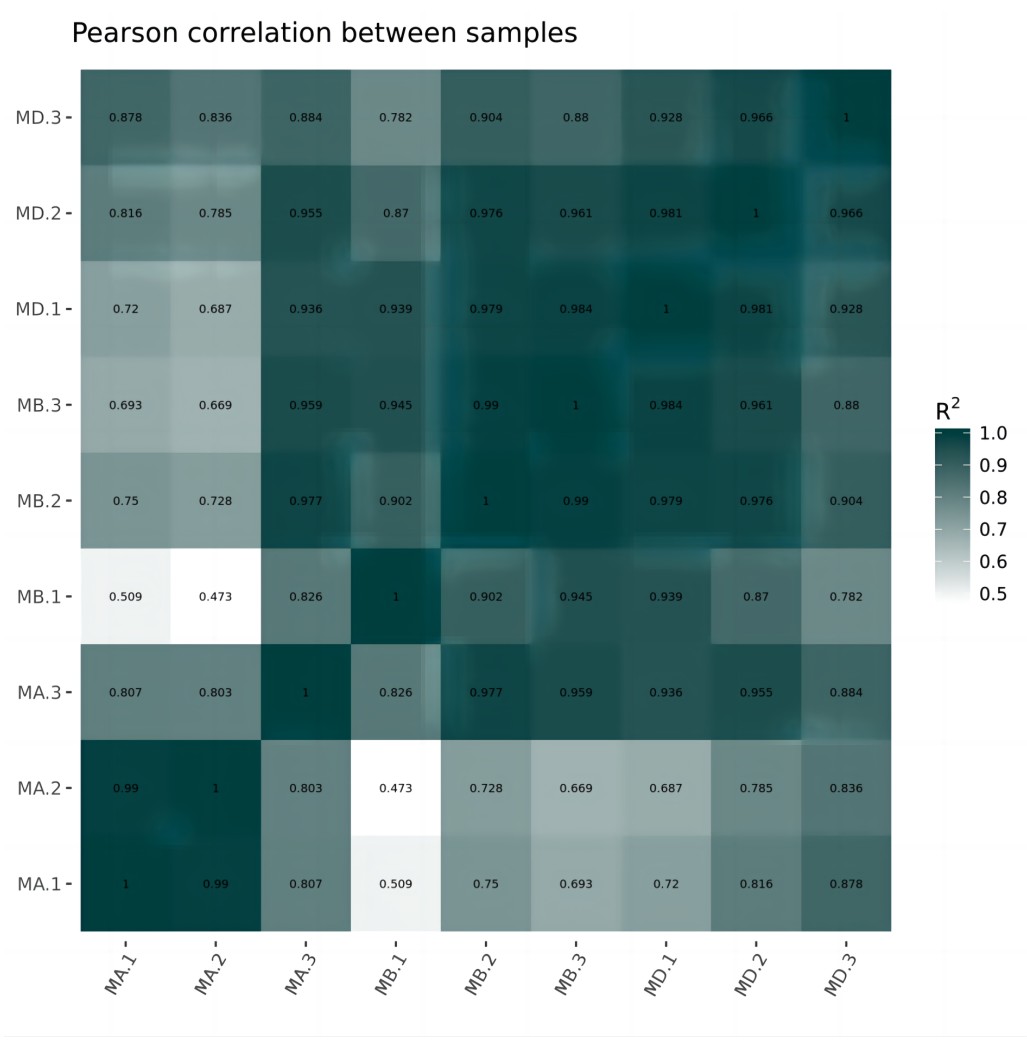

**Figure 2 Horizontal hierarchical clustering of different samples.** The same kind of samples (Groups MA, MB and MD) can appear in the same cluster. The genes clustered in the same color indicated the need for further data analysis.

Figures 4–9 illustrate the metabolites that are used for enrichment analysis and have been matched as well as the metabolites that were only used as background information and have not been matched. The related genes in the pathway analysis are associated in the actual signal pathway where they interact. Compared with GO analysis, pathway analysis is a more direct way to study differential genes. The purpose of signal pathway analysis was to find the signal pathway with significant difference gene enrichment based on the KEGG database. The selected differential genes were annotated to get all the pathway terms involved in the differential genes and target genes. The Fisher's test was used to calculate the significance level ($p$-value) of the pathway, to select the significant pathway terms enriched by the differential genes and negative correlation genes.
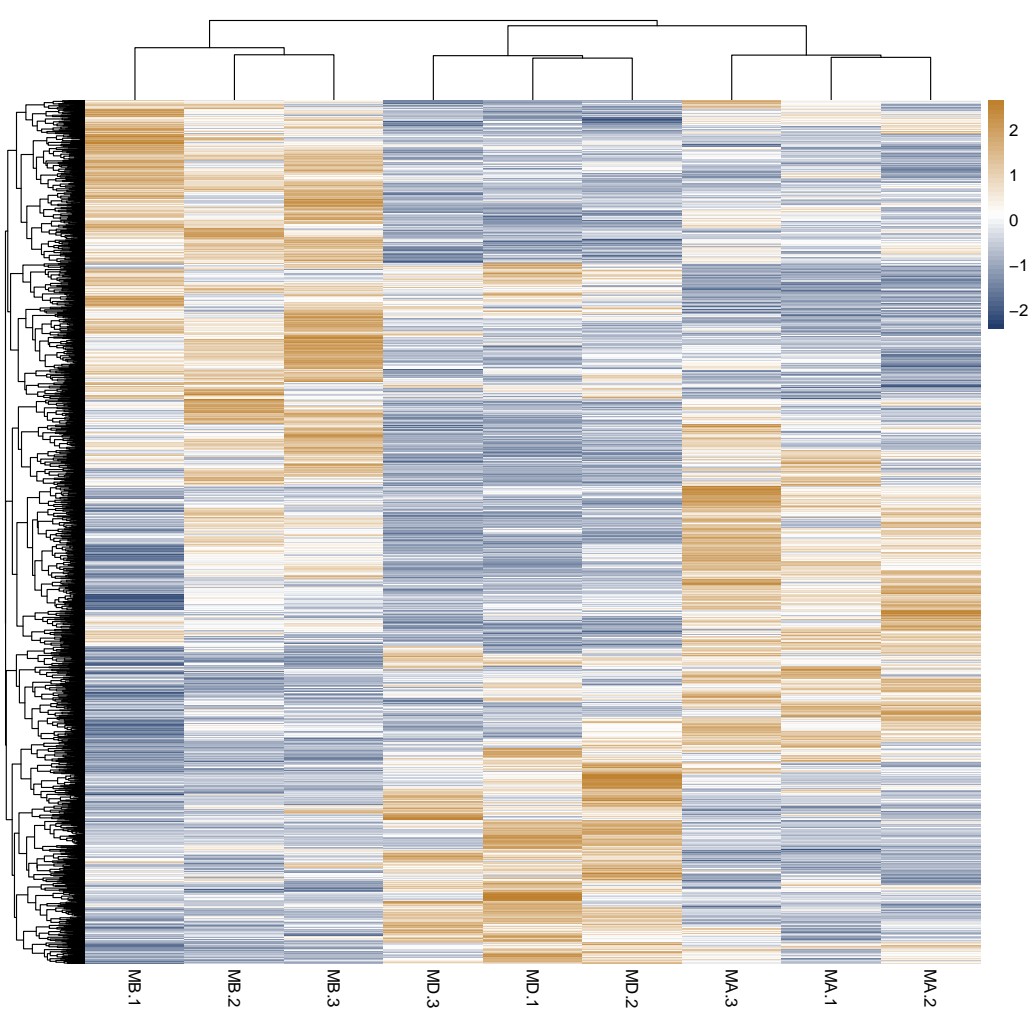

**Figure 3   Cluster analysis of different genes among of Groups MB, MD and MA.** The genes expression in the same group (Groups MA, MB and MD) was grouped. There was significant difference in the gene's expression among of groups MB, MD and MA, and the repeatability of the same group was good. There were significant differences in the genes expression of Group MB compared to MA, Group MD compared to MA and Group MD compared to MB.

Heatmap cluster analysis, GO, and pathway analysis were conducted among groups MB, MD, and MA to elucidate the differential expression of genes. A significant difference in mRNA expression was observed in the low concentration treatment group (Group MB) compared to the control group (Group MA), as depicted in Fig. 3A. The functional analysis of the differentially expressed genes showed significant up-regulation or down-regulation, as illustrated in Fig. 4, while the metabolic pathway analysis revealed similar trends, as shown in Fig. 5. Similarly, a significant difference in mRNA expression was noted in the high concentration treatment group (Group MD) compared to the control group (Group MA), as presented in Fig. 3B. The functional and metabolic pathway analyses of these genes are displayed in Figs. 6 and 7, respectively. Furthermore, significant differences in

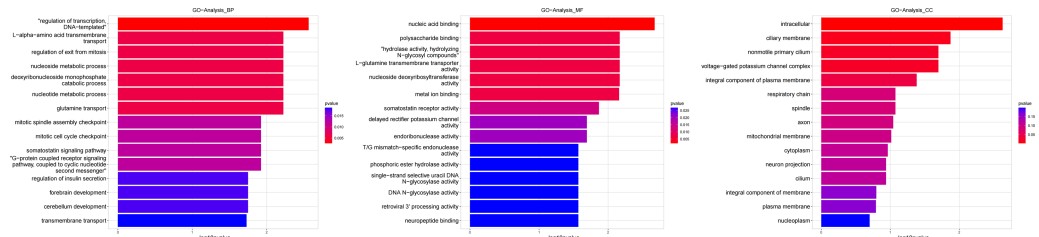

**Figure 4  Gene ontology (GO) analysis of biological process and molecular function between Groups MB and MA.** These significantly different expression genes between Group MB and Group MA were further analysis of the crucial genes related to male offspring mouse reproductive function by prochloraz's invasion response.

mRNA expression were observed in the high concentration treatment group (Group MD) compared to the low concentration group (Group MB), as shown in Fig. 3C. The functional and metabolic pathway analyses for these differentially expressed genes are depicted in Figs. 8 and 9, respectively.

## Genes function of mainly different expression among different groups

Ten up or down-regulated genes between two groups (B *vs* A, D *vs* A, and D *vs* B) were analyzed. The analysis required that the absolute value of the log2FC is greater than 0.585 and the *P* value be less than 0.01. From the initial set of 60 significantly differentially expressed genes, the same genes expressed in all three groups were selected for further analysis. Further analysis revealed that the expression of *Cwc22* and *Rsph3b* was significantly higher in both the control group and the high concentration group compared to the low concentration group, with no significant difference between the first two groups (as shown in Supplemental Information 3). The expression of the *Mt-nd6* (Mitochondrially Encoded *Nadh* Ubiquinone Oxidoreductase Core Subunit 6) gene was significantly higher in the control group compared to the low concentration group and the high dosage group, with no significant difference between the latter two groups (as shown in Supplemental Information 3). The expression of the *Slc12a4* (Solute Carrier Family 12 Member 4) gene in the experimental group (low concentration and high concentration groups) was significantly higher than in the control group, with no significant difference between the first two groups (as shown in Supplemental Information 3). *Dynlt1a, Olfr112, and Tpm3-rs7* were significantly higher in both the control group and the low concentration group compared to the high concentration group, with no significant difference between the latter two groups. *Fam124a* was significantly higher in the high concentration group compared to the control group and the low concentration group, with no significant difference between the latter two groups (as shown in Supplemental Information 3). The results showed that eight significantly expressed RNA genes (*Cwc22, Rsph3b, mt-Nd6, Slc12a4, Dynlt1a, Olfr112, Tpm3-rs7*, and *Fam124a*) in these three groups are mainly involved in the function of the mouse reproductive system through the mitochondrial function pathway, as shown in Table 2.
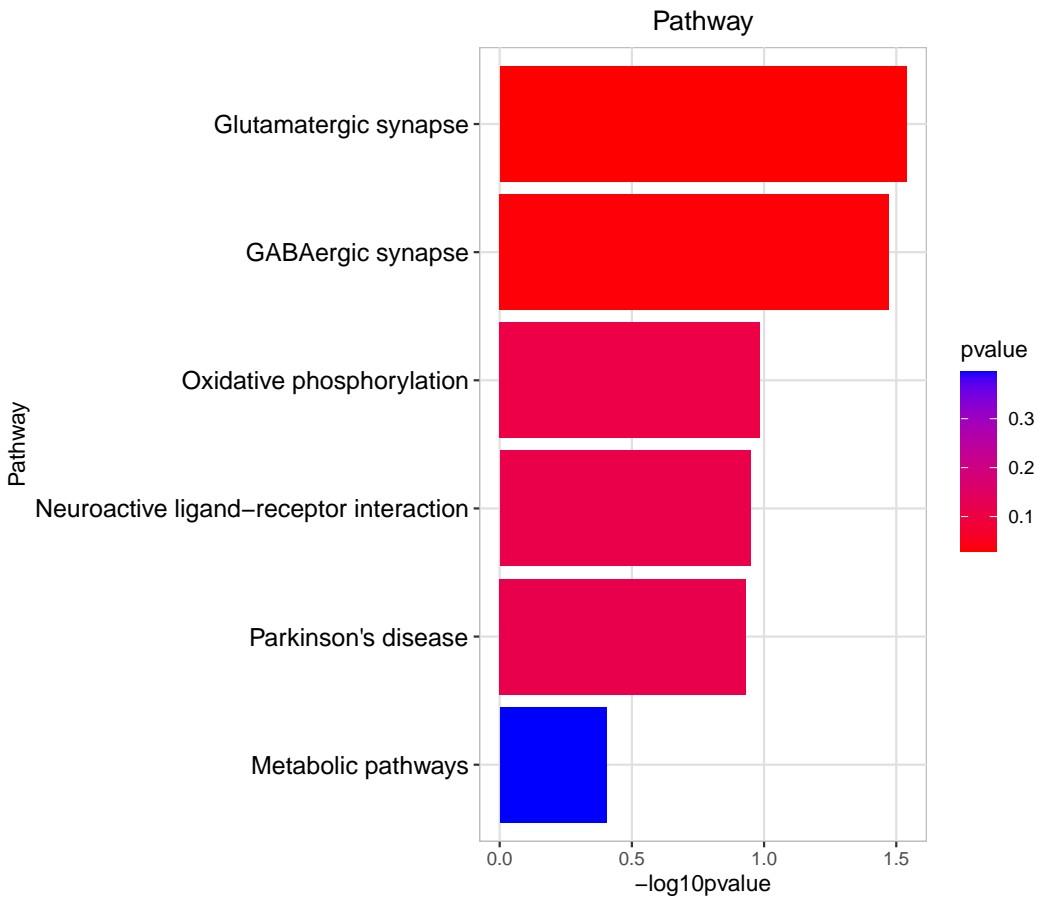

**Figure 5** **Pathway analysis between Groups MB and MA.** The pathway analysis shows that energy metabolism and oxidative responses were impacted by the mitochondrial response to sperm motility by the prochloraz invasion response.

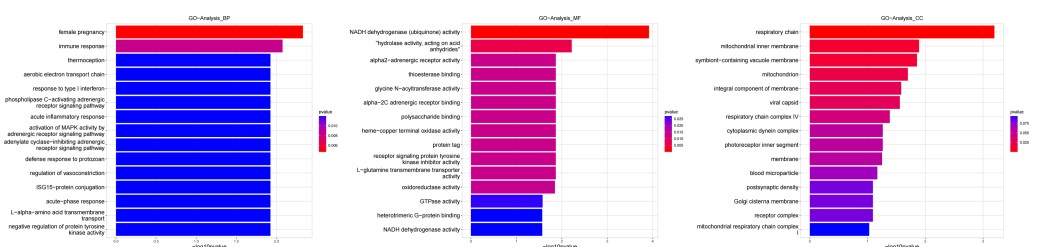

**Figure 6** **Gene ontology (GO) analysis of biological process and molecular function between Groups MD and MA.** These significantly different expression genes between Group MD and Group MA were further analysis of the crucial genes related to the reproductive function of male offspring and the prochloraz invasion response compared to the control.

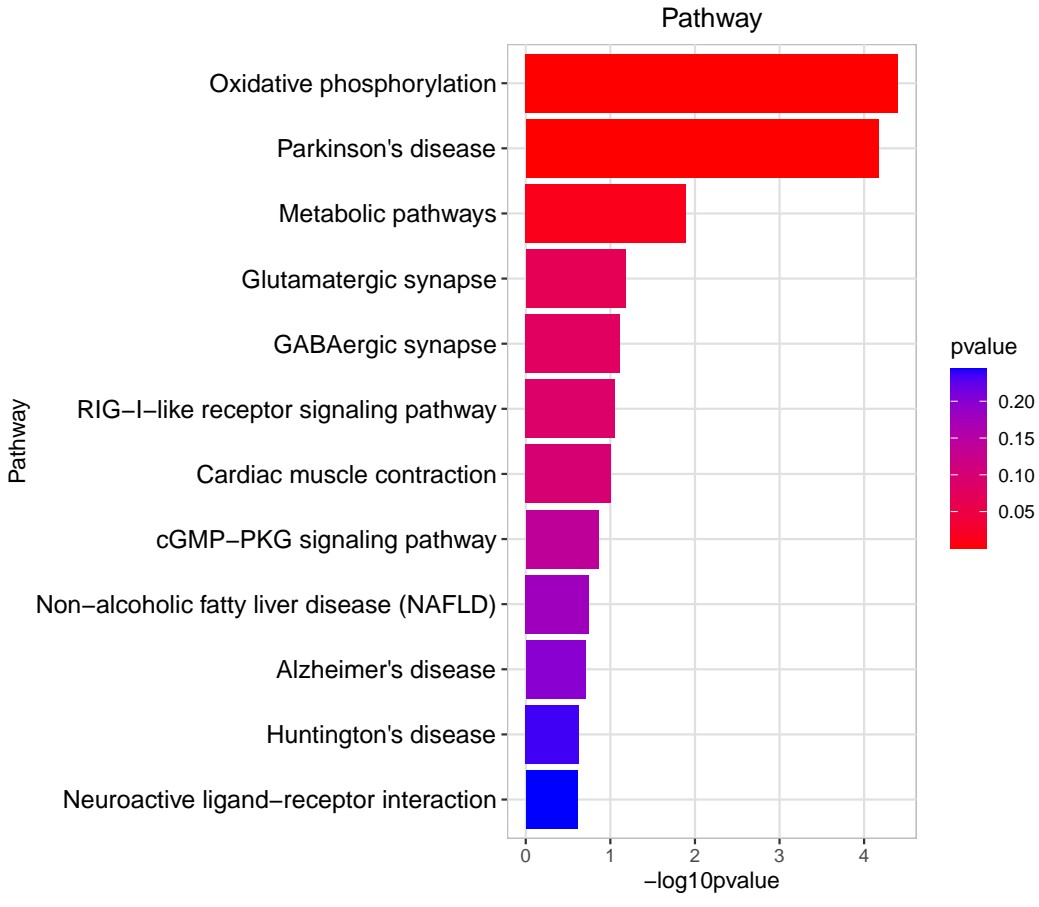

**Figure 7 Pathway analysis between Groups MD and MA.** The pathway analysis mainly involved energy metabolism and the oxidative response by mitochondria for sperm motility in male mice by the prochloraz invasion response compared to the control.



**Figure 8 Gene ontology (GO) analysis of biological processes and molecular function between Groups MD and MB.** These significantly different expression genes between Group MD and Group MB were further analysis of the crucial genes related to male offspring mouse reproductive function impacted by prochloraz's invasion response.

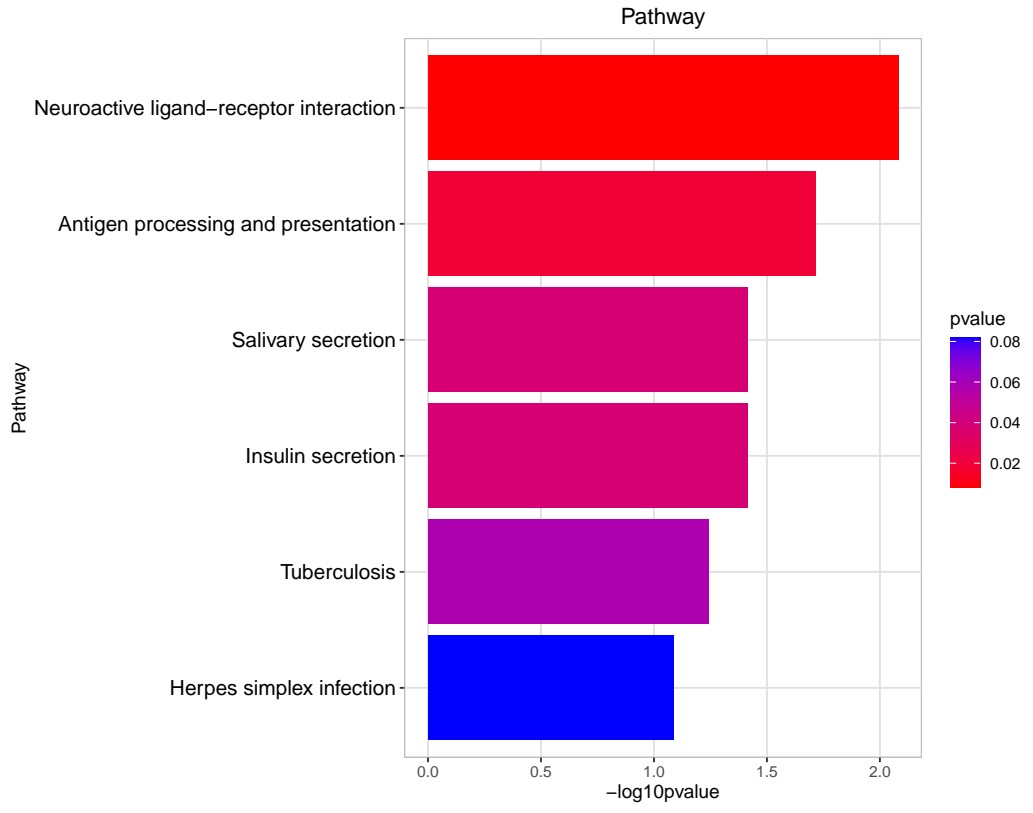

**Figure 9  Pathway analysis between Groups MD and MB.** The pathway analysis involved a neuroactive response by estrogenic hormones for male mouse sperm development impacted by a prochloraz invasion response between Group MD and Group MB.

Further analysis found that selected RNAs (*Greb1, Esrrb, Catsperb, Mospd2, Sohlh1* and *Specc1*) were closely related to sperm functions and estrogen reaction, which is significantly expressed among different groups (as shown Table 3).

## DISCUSSION

It has been suggested that exogenous chemical exposure will affect the endocrine and reproductive systems of the mouse during critical developmental periods in pregnancy (*Lim et al., 2018*). Prochloraz, a widely used pesticide in global agriculture production, is known to possess anti-androgenic and estrogenic properties that may be harmful to human reproductive health. Despite numerous reports on the adverse effects of prochloraz usage, there is a lack of systematic studies on the transcriptional changes at the mRNA molecular level in the reproductive system of male mouse offspring. This study involved injecting male father mice with varying dosages of prochloraz for two weeks and found that it led to a reduction in the growth of male mouse reproductive organs, as evidenced by decreased testis weight and testicular coefficient index in male offspring mice (Table 1). These findings align with previous research (*Lundqvist, Hellman & Oskarsson, 2016*; *Vinggaard et al., 2005*).

Table 2  The eight significantly different expression genes among three different groups. These genes indicate that the absolute value of the log2FC be greater than 0.585 and the $P$ value be less than 0.05.

| Gene name | Group B VS A | | Group D VS B | | Group D VS A | |
|---|---|---|---|---|---|---|
| | Value of log2FC | P | Value of log2FC | P | Value of log2FC | P |
| Cwc22 | −1.990304662 | 1.05E−02 | 1.781116721 | 1.54E−02 | | |
| Rsph3b | −1.603312422 | 7.09E−22 | 1.656498331 | 2.86E−25 | | |
| mt-Nd6 | −1.266568432 | 1.03E−02 | | | −1.632540494 | 3.57E−04 |
| Slc12a4 | 1.672896772 | 3.80E−02 | | | 2.10288627 | 1.15E−48 |
| Dynlt1a | | | −1.916787646 | 3.92E−03 | −2.060585715 | 2.70E−03 |
| Olfr112 | | | −1.830378862 | 4.60E−14 | −1.670733055 | 3.39E−19 |
| Tpm3-rs7 | | | −1.728976432 | 2.10E−03 | −1.46052739 | 2.02E−02 |
| Fam124a | | | 1.814625218 | 8.60E−08 | 1.972226535 | 1.32E−04 |

Table 3  The different significantly up or down expressed genes related to sperm functions and estrogen reaction among different groups. These genes were selected related to sperm functions and estrogen reaction among different groups

| Gene name | Group B VS A | | Group D VS B | | Group D VS A | |
|---|---|---|---|---|---|---|
| | log2FC | P | log2FC | P | log2FC | P |
| Greb1 | 0.690200666590 | 0.017344567 | 0.790416254 | 0.002541977 | 1.474231586 | 1.32E-14 |
| Catsperb | −1.267997125 | 0.00288627 | 1.101498307 | 4.60E-09 | | |
| Mospd2 | −0.629304332 | 0.026731419 | | | −0.667487059 | 0.007417209 |
| Sohlh1 | 0.939524273 | 0.001194771 | −0.62972177 | 0.024680643 | No significant difference | |
| Esrrb | 0.851861356873 | 0.000693787 | −0.832231705 | 0.002510378 | | |
| Specc1 | No significant difference | | | | 0.607947913 | 0.000245066 |

In our study, we initially observed that the Pearson correlation coefficient distance clustered into three groups based on sequencing data from three samples in each group. Analysis of these raw data sets, as depicted in Supplemental Informations 1, 2, and Fig. 1, revealed that while the repeatability of the same sample was high, there were significant differences between samples subjected to different treatments (*Kim et al., 2018*). Subsequent clustering analysis was performed to elucidate variations in gene expression among different samples, demonstrating that samples within the same group exhibited clustering with similar biological significance (as illustrated in Fig. 3 for groups MB *vs* MA, MD *vs* MA, and MD *vs* MB). Building upon this validated sequencing data, we conducted an analysis of ten significantly up- or down-regulated RNA genes to explore the relationship between mRNA expression clusters and changes in mouse reproductive function induced by varying dosages of prochloraz.

This study identified differences in expression levels of selected genes (Cwc22, Rsph3b, Mt-nd6, Slc12a4, Dynlt1a, Olfr112, Tpm3-rs7, and Fam124a; as listed in Supplemental Information 3 and Table 2) among various experimental groups compared to the control group. Specifically, Mt-nd6 and Slc12a4 exhibited significant differences in expression between the experimental groups and the control group, indicating a potential relationship between prochloraz effects on mouse male testis development and the mitochondrial function pathway, particularly involving these genes. Notably, Mt-nd6 and Slc12a4 are

associated with sperm development and fertilization with oocytes in female reproductive tubes by influencing mobility (*Moser et al., 2009*; *Thomas & Dong, 2019*). These mRNAs are also closely linked to mouse male reproductive system development, as previously reported. The analysis revealed that Mt-nd6 gene expression was low in the experimental group, while Slc12a4 gene expression was high in the experimental group, especially in the low and high concentration prochloraz experimental groups, with no significant difference between the two concentrations (as shown in Supplemental Information 3 and Table 2). Previous research has suggested that Mt-nd6 plays a crucial role in Complex I function, responsible for electron transfer in the respiratory chain and sperm mitochondrial function (*Mao et al., 2020*). It has been observed that mutations, deletions, and copy number variations in mitochondrial genes are associated with sperm activity (*Mao et al., 2020*). Research has also indicated that mitochondria in the cytoplasm are important targets for estrogenic effects. The inner mitochondrial membrane plays a crucial role in converting cholesterol into pregnenolone, a precursor to various steroid hormones such as androgens and estrogens (*Kemper et al., 2013*; *Wu et al., 2021*). Additionally, studies have reported that all key steps in the synthesis of steroid hormones take place in mitochondria within steroid-producing tissues (*Picard, 2022*). The expression level of Mt-nd6, a mitochondrial inner membrane protein involved in NADH production, can serve as an indicator for evaluating cellular metabolic activity and mitochondrial function, potentially impacting sperm function. Furthermore, Slc12a4 is primarily involved in mediating electroneutral potassium-chloride cotransport, particularly in response to cell swelling, which aids in maintaining cell volume homeostasis (*Klein, Cooper & Yeung, 2006*; *Moser et al., 2009*). Sperm must undergo regulated volume decrease (RVD) post-ejaculation to counteract swelling caused by low osmotic pressure in the female urethra, and defects in sperm RVD can lead to failure in connecting with the mouse uterus fallopian tube (*Moser et al., 2009*). In conclusion, the Mt-nd6 and Slc12a4 genes are key regulatory genes of prochloraz that influence mouse reproductive system function through mitochondrial information processing, serving as the molecular mechanism by which prochloraz affects reproductive system function. Moreover, analysis has revealed that specific RNAs (Greb1, Esrrb, Catsperb, Mospd2, Sohlh1, and Specc1) are closely linked to sperm functions and estrogen responses, showing significant expression variations across different groups (refer to Table 3).

The reports found that the *Greb1* (gene regulated by estrogen in breast cancer protein) and Esrrb (estrogen related receptor, beta) is closely relative to estrogen stimulation. These studies show that *Catsperb, Mospd2, Sohlh1* and *Specc1* are all closely related to the motility of sperm (*Amini et al., 2014*; *Di et al., 2018*; *Guo et al., 2015*; *Jaiswal et al., 2014*; *Lee & Choi, 2021*). It has been observed that only two genes, *Greb1* and *Mospd2*, show significant differences in expression between the two experimental groups and the control group. This suggests that the effects of prochloraz on male testis development in mice may be related to the functions of these genes, similar to estrogen and sperm function (as shown in Table 3).

The gene numbers and their significance in regulating DNA transcription and transmembrane transport pathways are depicted in Figs. 4–5, which show the significantly

differentially expressed genes identified for cluster function analysis in Group MB compared to Group MA (the control group). It is well-established that biological organisms typically enhance protein synthesis and cell transmembrane transport in response to exposure to toxic substances to facilitate rapid detoxification. Additionally, it is known that DNA transcription activity is particularly robust during processes related to toxicological responses in the mouse reproductive system (*Svingen et al., 2018*). Upon administration of higher doses of prochloraz, many significantly different genes in Group MD (higher dosage group) were associated with female pregnancy and immune responses (Figs. 6–7) compared to Group MA (the control group). This finding is consistent with previous studies that have reported prochloraz's estrogen-like effects on the male and female reproductive systems (*Melching-Kollmuss et al., 2017*; *Vinggaard et al., 2005*; *Vinggaard et al., 2006*). A comparison between Group MD and MB reveals significant changes in protein complex assembly, neurotransmitter secretion, neuropeptide signaling pathways, and immune response (Figs. 8–9), suggesting a potential role in the early development of the male reproductive system. In summary, this study offers novel insights into global transcriptome changes and the abundance of specific transcripts in the mouse male testis following exposure to varying doses of prochloraz during growth. The findings indicate that the functions of many genes induced by prochloraz stimulation are primarily focused on male reproductive development through hormone production and immune response.

Although these genes were predicted through our GO and pathway analyses, their associations are limited, which has also been verified in other studies. We identified differentially expressed mRNAs from male mouse testis exposed to varying dosages of prochloraz (groups MA, MB, and MD). Our GO and pathway analyses revealed a cluster of differentially expressed genes closely linked to mitochondrial function, and sperm motility and development, regulated by endocrine hormones such as estrogen (Greb1, Esrrb, Catsperb, Mospd2, Sohlh1, and Specc1), which are crucial for the development of the male mouse reproductive system. These genes could potentially serve as biomarkers to deepen our understanding of how prochloraz affects the male reproductive system, opening avenues for further investigation. Nevertheless, there are still numerous unanswered questions concerning the mechanisms behind the potential impact of estrogen-like pesticides on male reproductive epigenetic function. Furthermore, it remains uncertain whether exposure to prochloraz influences the expression patterns of related genes in male offspring mice through inter-generational genetic effects.

### Funding

This work was supported by the Scientific Research Project of Hunan province Education Department (20A280) and the Hunan Provincial Natural Science Foundation of China (2024JJ7241). The funders had no role in study design, data collection and analysis, decision to publish, or preparation of the manuscript.

## Grant Disclosures

The following grant information was disclosed by the authors:
Scientific Research Project of Hunan province Education Department: 20A280.
Hunan Provincial Natural Science Foundation of China: 2024JJ7241.

## Competing Interests

The authors declare there are no competing interests.

## Author Contributions

- Junhe Hu conceived and designed the experiments, performed the experiments, prepared figures and/or tables, authored or reviewed drafts of the article, and approved the final draft.
- Chang Liu conceived and designed the experiments, performed the experiments, prepared figures and/or tables, authored or reviewed drafts of the article, and approved the final draft.
- Xianghui Zeng conceived and designed the experiments, performed the experiments, prepared figures and/or tables, and approved the final draft.
- Tao Tang conceived and designed the experiments, performed the experiments, prepared figures and/or tables, authored or reviewed drafts of the article, and approved the final draft.
- Zhi Zeng analyzed the data, prepared figures and/or tables, and approved the final draft.
- Juan Wu analyzed the data, prepared figures and/or tables, and approved the final draft.
- Xiansheng Tan analyzed the data, prepared figures and/or tables, and approved the final draft.
- Qingxiang Dai analyzed the data, prepared figures and/or tables, and approved the final draft.
- Chenzhong Jin analyzed the data, prepared figures and/or tables, and approved the final draft.

## Animal Ethics

The following information was supplied relating to ethical approvals (i.e., approving body and any reference numbers):

The study is approved by the Institutional Ethics Committee of Hunan University of Humanities, Science and Technology (20211010).

## Data Availability

Data is available at the National Genomics Data Center: CRA012977.

## Supplemental Information

Supplemental information for this article can be found online at http://dx.doi.org/10.7717/peerj.17917#supplemental-information.

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
