# Peer review of "Prochloraz induced alterations in the expression of mRNA in the reproductive system of male offspring mice"

_PeerJ, doi:10.7717/peerj.17917_

## Round 0.1 · original submission · Major Revisions

The manuscript has been assessed by two independent reviewers and I strongly suggest addressing the concerns raised by the reviewers before your paper can be considered for publication.

Language revision is highly recommended.

Abstract needs restructuring for better understanding of the rationale behind the current study.

Experimental design and figures needs extensive revision.

Authors are suggested to provide more details on the sample collection and tools used.

More concentration is required on the presentation of the results.

**Language Note:** The Academic Editor has identified that the English language must be improved. PeerJ can provide language editing services - please contact us at [email protected] for pricing (be sure to provide your manuscript number and title). Alternatively, you should make your own arrangements to improve the language quality and provide details in your response letter. – PeerJ Staff

Reviewer 1 ·

Basic reporting

The manuscript should be reorganized before judging. The following comments probably will be helpful.

1. Double check the body weight of 3 or 4-week-old mice (line 79-80).
2. 3-week-old mice is not perinatal stage (line 19).
3. The group name is confused. What are the full names of “MA”, “MB” and “MD”?
4. Double check the mating ratio (line 111).
5. The description on RNA-seq quality control and mapping should be reduced or removed (line 179-207), and focus more on the downstream analysis and validation.
6. On line 212-215, the description is not correct, all the methods have both FP and FN. Suggest to remove it.
7. The description on the heatmaps is not attractive and the details are missing (line 225-234).
8. “GO” vs “Go” vs “go”?
9. Check the gene nomenclature. Gene symbols generally are italicised, with only the first letter in uppercase and the remaining letters in lowercase.
10. “selected mRNAs” should be “selected genes” (line 319).
11. Ref missing after “previously reported” (line 328).
12. On line 377, mRNA genes??
13. On line 385, these miRNAs???
14. Move the figure legends at the bottom of figures.
15. Why the author not group the heatmaps (Figs. 2-4)?
16. The authors have to point out the colors’ meanings (blue and red) in Figure 5-8 and 10.
17. The authors have to indicate how to calculate “testicular coefficient index”.
18. Regarding the phenotype of drug treated male mice, how about the phenotype of the sperm? How about the testicular histology?
19. The identified DEGs should be validated.

Experimental design

OK

Validity of the findings

Yes

·

Basic reporting

The manuscript by Hu et al. described the alteration in mRNA expression of male mice in response to the fungicide prochloraz. The authors report that exposure to prochloraz reduces testicular weight and alters Mt-nd6 and Slc12a4 mRNA expression. The following points should be addressed before acceptance.
1. The English language should be improved to ensure an international audience can clearly understand your text. The phrasing and grammar used in the manuscript are frequently incorrect. Phrases like that in lines 197/198 “At present, there are many mapping software in the world” are redundant. Lines 323 and 358 imply your results are guesswork and should be rephrased. The reporting of results in Section 3.6 is difficult to understand. I recommend restructuring or rewriting this section to improve the clarity of the results. In line 280 you refer to the “high difficulty group”, this should be rephrased to avoid ambiguity.
2. The abstract is ambiguous and does not inform the reader of your results. Reporting of the experimental groups would be clearer if they were referred to as the concentration of prochloraz e.g. 0 mg/kg, 53.33 mg/kg or 160 mg/kg rather than MA, MB or MD. Reporting one of your major findings was that RNA-Seq data was mapped to the genome (lines 24-25) isn’t necessary.
3. Figures 2-4 are prepared in red and green. I recommend these figures be redesigned in a pallet so that people with colour blindness can visualise the results. Currently, this poses an issue for the inclusivity of the manuscript to the audience. Additionally, these figures should be annotated more clearly. You could also consider combining the figures.
4. Figured 5-10 should be improved. In many instances, text quality is low and cannot be read. Full labels should be added as several gene names are cut off. The decision to have bars in red vs blue should be explained.
5. In lines 278-284 it is reported that Mt-nd6 expression is higher in control groups and slc12a4 was higher in exposed groups as reported in S2. Upon review of S2, I have not been able to determine how this conclusion has been drawn. The raw data should be uploaded upon resubmission for review.
6. The impact of prochloraz on mRNA expression is unclear. You describe that a number of mRNAs expression is altered (table 2) but report that only 2 (Mt-nd6 and Slc12a4) have significant differences from the control (line 323). Please clarify what this means and improve the presentation of the results (S3 and Table 2) so it is immediately apparent what the results are.
7. Lines 350-352 report results from Table 4, it is unclear if results have been omitted or if it is meant to direct the reader to Table 3.
8. Reporting of results in Table 2 as “mainly significant” is ambiguous and should be clarified.
9. “GO” should be defined. You should also be consistent with how it is referred to; GO, go and Go are all used.
10. The statistical significance of the results reported in Table 1 is difficult to interpret. The letters should be defined in the Table legend.

Experimental design

Addressed in the basic reporting section.

Validity of the findings

Addressed in the basic reporting section.

---

## Round 0.2 · Major Revisions

The manuscript has undergone thorough evaluation by two reviewers. While both reviewers have provided valuable feedback, they have raised some major concerns and minor points that need to be addressed for your manuscript to be considered for publication.

1. Language has been improved, however, further revisions are still necessary to ensure its overall clarity and coherence.

2. Please address discrepancies in mouse weight and provide clarity in heatmap interpretation.

3. Correct instances of "mRNA genes" and ensure experimental evidence supports your findings.

4.Clarify unclear statements and provide more explicit conclusions from your GO analysis.

5. Remove irrelevant lines (194-202) that do not contribute to the results.

6. Please attend to minor corrections in your manuscript, such as providing full names for DEHP and DBP, including missing references, using appropriate terminologies and ensuring clarity in figure legends.

7. Testicular weight reporting is unclear, and potentially contradictory. Revision is needed for accuracy and clarity.

Reviewer 1 ·

Basic reporting

See additional comments

Experimental design

See additional comments

Validity of the findings

See additional comments

Additional comments

This version of manuscript is much improved. However, I still have following major concerns:
1. Thank you for responding to my comments, but some of them still unclear.
1.1. Thanks authors for clarifying the body weight of mice. However, based on your current writing (Line 83 and 84), my understanding is that 3wk-old male mice weigh 25+/-2g and 4wk-old female mice weigh 20-22g. Is it correct?
1.2. Thanks authors to group the heatmaps to a figure. However, the present pairwise analysis may obscure the expression patterns of DEG within the third group. For instance, in Fig. 2A, the expression of DEGs in MD group remains ambiguous? It is recommended that the authors create a unified heatmap incorporating all three groups (MA, MB and MD). Additionally, clarification is needed regarding the criteria used to determine “significant difference” and “repeatability” in the description of the heatmap in the Results section?
1.3. Some “mRNA genes” still not corrected. For example, line 263 and 266.
1.4. I find your reply to my last comment (#19) unclear. Since the RNA-seq analysis presented in this manuscript appears straightforward, it is crucial to include experimental evidence to support the findings.
2. Some writing remains ambiguous. For instance, it is unclear what the authors want to say in lines 248 and 260? Furthermore, what conclusions can be drawn from your GO analysis?
3. Remove lines 194-202. They are not your results.

Other minor points
1. What are the full names of DEHP and DBP on line 46?
2. Line 52, “Ah receptor”, provide the full name of “Ah”.
3. Reference is missing on line 50 after “South America”.
4. Rewrite the sentence of line 70 to 72. Hard to understand.
5. Provide Cat# of your Trizol Reagent on line 129.
6. Lowercase “The” on line 148.
7. Line 196, what is “epibiotek”? A name of company?
8. “highconcentration” on line 219.
9. What are “grape genes” on line 231?
10. Remove extra “e” on line 342.
11. “showsignificant” on line 353.
12. Figure 1 legend, what is the “cluster t”?
13. Figure 2 legend, not clear. “the expression of gene expression”?? “among group MB, MD and MA” should be “among groups MB, MD and MA”, “the repeatability of the same sample is good”, I think the authors want to say “the same group”.
14. Figure 3 and 5 legends. Appeared some non-English words. Some of the GO term name are uncomplete in Figures 3-8.

·

Basic reporting

While there has been an improvement in the English language many grammatical errors remain throughout. In the abstract alone, this includes a space before the full stop in line 17 and a double space in line 19. Figure legends 3 and 5 have numerous errors and include unusual characters. Overall the English should be addressed further.

The results reported in line 20 do not make sense to me. You state that testicular weight is lower in the treatment groups (MB and MD) than the control group (MA), yet the results are reported as 0.312, 0.294, and 0.355 g. To me, this appears that the MD group had the largest weight. Please rephrase this to avoid ambiguity.

Points 3 and 4 have not been addressed despite the authors claiming to have taken onboard the suggestions. I strongly suggest that red and green be avoided to ensure people with colour blindness will be able to interpret the results. Text on many of the graphs remains too small to read. Text on Figures 3, 4, 5, 6, 7 and 8 is cut off on the x-axis. Figure quality should be improved before publication.

The use of the phrase ‘mRNA genes’ is inaccurate and needs to be addressed as pointed out by reviewer 1.

Experimental design

Addressed in the basic reporting section.

Validity of the findings

Addressed in the basic reporting section.

---

## Round 0.3 · Minor Revisions

I appreciate the improvements made in this version of the manuscript, however, there are still some remaining concerns:

1. The text in Figures 3, 5, and 7 still remains too small, hindering readability.

2. The term "mRNA gene" persists in several instances throughout the manuscript, including lines 273 and 300, as well as in Table 2.

3. The legends of Figure 3 still contain unusual characters that require rewording for clarity.

4.Authors are suggested to clarify whether the color scale bar in the heatmap corresponds to Z-scores? Clarification on this aspect would aid in comprehending the interpretation of the heatmap data.

Addressing these issues will further enhance the quality and clarity of the manuscript, ensuring its effectiveness in conveying the research findings to readers. Hence, I strongly suggest the authors to address the comments from the reviewers to be considered for publication.

Reviewer 1 ·

Basic reporting

see Additional Comments

Experimental design

see Additional Comments

Validity of the findings

see Additional Comments

Additional comments

Thanks authors to address all my comments.
One more minor point:
What is the color scale bar in heat-map represents? Z-score?

·

Basic reporting

This version of the manuscript is much improved, thank you for addressing my comments. The presentation of Figure 2 is much improved in a different colour palette. However, I still have the following comments
- Text in Figures 3, 5 and 7 remain too small to be read.
- 'mRNA gene' remains in lines 273 and 300, and in Table 2
- The legends of Figure 3, still contain unusual characters.

Experimental design

See basic reporting

Validity of the findings

See basic reporting

Additional comments

See basic reporting

---

## Round 0.4 · Minor Revisions

I appreciate author's effort for overall improvement of the manuscript, however, there are still a few concerns that need attention for further improvement.

1. I strongly recommend including a schematic diagram outlining the experimental protocol to enhance clarity for readers, especially considering the transgenerational nature of the study.
2. There are numerous typos and spacing errors throughout the manuscript, such as in Figure 3 legend. I suggest thorough proofreading to rectify these issues.
3. Furthermore, the discussion section could be enhanced by delving deeper into the pathways implicated by the highlighted genes, particularly regarding paternal mitochondria breakdown post-fertilization.
4. Lastly, incorporating proper figures, such as testes size comparisons and histological images, would enrich the manuscript. I believe addressing these points would significantly strengthen the manuscript.

---

## Round 0.5 · Minor Revisions

I appreciate the author's effort in including the schematic diagram and overall improvements to the manuscript. However, there are a few concerns that need attention for further enhancement. I strongly encourage the authors to incorporate histological images of the testes to enrich the manuscript. Additionally, I suggest providing evidence of editing by a fluent English speaker or a professional service.

---

## Round 0.6 · accepted · Accept

The authors have addressed the suggested revisions, have significantly improved the quality of the manuscript and I recommend it for publication.